

**The inference of internal solitary waves in the northern South China Sea from**
**data acquired by underwater gliders**
Wei Ma[1,2], Hongwei Zhang[1], Chenyi Luo[1], Yanhui Wang[1,2], Yang Song[1]
[1] Key Laboratory of Mechanism Theory and Equipment Design of Ministry of Education, School of Mechanical Engineering, Tianjin
University, Tianjin, China
[2] Joint Laboratory of Ocean Observation and Exploration, Pilot National Laboratory for Marine Science and Technology, Qingdao,
Shandong, China
Correspondence to: Yanhui Wang (email: yanhuiwang@tju.edu.cn)
**Abstract.** Internal solitary waves (ISWs) are typical large-amplitude nonlinear waves occurring in stratified oceans.
The *in situ* observations of ISWs are needed to improve the regimes of nonlinear internal wave theories. There is
violent mixing of water mass in the horizontal and vertical directions during the propagation of ISWs, which generally
lasts for a short period at a fixed position. However, an underwater glider, with the features of low-speed and sawtooth
motion, cannot obtain a complete thermohaline stratification before and after the ISWs arrival. Those thermohaline
data collected *in situ* by gliders, which vary synchronously at spatial-temporal scales, raise challenges for identifying
the ISWs. Four Petrel-II gliders are deployed in the active region of ISWs in the South China Sea. This paper estimates
vertical water velocity from glider flight data and kinematic model, analyzes the sensitivity of parameters in the glider
kinematic model, and adopts a standard nonlinear search method to calibrate the parameters insensitive to the vertical
velocity. The depth-keeping experiment is performed to verify the effectiveness of the optimized results. The standard
deviation of vertical water velocity in the eastern Dongsha Atoll is revealed, and its distribution indirectly reflects
that the strength of vertical water activity increases gradually at the same latitude along the east-west direction. Using
observations of vertical water velocity fluctuations and isothermal surface vertical displacements, single- and
multiple-wave packets can be identified. The availability of this method is tested by comparison with a MODIS image.
Such an analysis provides a basis for the application of glider in the observation of ISWs.
**Keyword:** glider, vertical water mass velocity, single-wave packet, multiple-wave packet

**1. Introduction**
Internal solitary waves (ISWs) are ubiquitous features in the ocean, and they can propagate over thousands of
kilometers from the generation site with unusually strong currents, which may promote the exchange of nutrients and
biological production, and threaten the safety of ocean engineering or platforms (Cai et al., 2012; Simmons et al.,
2011; Shroyer et al., 2010). To well understand the ISWs, several theories have been proposed (Cai et al., 2014). The
*in situ* observation of ISWs contributes to the development of these theories. Major observational methods include 1)
remote sensing, which owns the characteristic of large-scale space (Zhao et al., 2004; Jackson, 2007), but with limited
capability in sensing the sea surface; 2) moorings, which can obtain high-resolution thermohaline and current
structures (Ramp et al., 2004; Alford et al., 2012), whereas conduct observation at a fixed position; 3) research vessels,
which can conduct the well-designed and targeted survey in combination with field data (Farmer et al., 2011; Liang




et al., 2019), only being time-consuming and costly.
In the previous decades, the autonomous platforms (such as float and glider) played a significant role in the
observation of some essential ocean variables. As an important member, gliders have been widely used in the
oceanographic measurements (Whitt et al., 2020; Rudnick, 2016; Testor et al., 2019). The glider is driven by a
variable buoyancy engine to sink and rise alternately between the surface and a depth of 1000m or more. With the
aid of wings, the glider flies along a sawtooth trajectory with the glide speed of 0.25m/s or horizontal distance of
~20km/day. The underwater attitude of glider is adjusted by shifting or rotating the eccentric battery pack (or rudder)
under the navigation of electronic compass. Upon surfacing, the glider performs positioning via GPS, communicates
with shore-based control center and transmits part of observational data via satellite. The low-power consumption
and low-speed cruising enable glider to conduct long endurance missions, which can last several months or up to a
year at temporal scales and span several hundreds or even thousands of kilometers on spatial scales.
The observation of internal waves or tides conducted by gliders has been reported in several studies (Rudnick
et al., 2013; Boettger et al., 2015; Johnston and Rudnick, 2015; Johnston et al., 2015; Todd, 2017; Hall et al., 2017;
Ma et al., 2018; Hall et al., 2019). As a type of nonlinear internal waves, ISWs are very active in the South China Sea
(SCS), which exist with amplitudes up to 100m and phase speeds of 0.7~2.9m/s (Cai et al., 2012). The depression
wave and elevation wave have different structures and cause contrasting vertical fluctuations of seawater along the
direction of wave propagation. The ISWs observed in the SCS could be classified into two categories: a single-wave
packet, which contains a single ISW with/without an oscillating tail, and a multiple-wave packet composed of a group
of rank-ordered ISWs. According to the *in situ* observation, a conventional solution to identify and classify the waves
is to analyze the displacement of isopycnals and isotherms or the fluctuation of currents (Ramp et al., 2004; Ramp et
al., 2010; Huang et al., 2016). However, the motion characteristics of glider determine that it cannot achieve the
fixed-station observation as moorings to obtain complete thermohaline stratification at different depths before and
after ISWs arrival. It is difficulty in identifying and analyzing IWSs using glider observations which mix temporal
and spatial scales simultaneously. Therefore, it is questionable to analyze the vertical thermohaline fluctuations to
identify ISWs and classify the types of ISWs with the glider.
ISWs in the process of propagation induce strong currents simultaneously, involving horizontal and vertical
water velocities. Given the extra costs on the current sensors, the horizontal water velocity measured by gliders is not
discussed here. The observation of vertical water velocity by the near-neutral glider has been proven feasible in
practice (Merckelbach et al., 2010). The vertical velocity of water mass itself is the difference between the depth-rate
measured by pressure transducer from CTD and the glider's velocity through still water, while the latter can be
estimated from the kinematic model. With the vertical water velocities derived from the glider, the passage of internal
waves can be confirmed, and the intensity of internal waves in the survey area can be mapped (Rudnick et al., 2013;
Todd, 2017). Therefore, the vertical water velocity, combined with high-resolution hydrographic data may be a
solution to reflect the feature of ISWs, which is presented and verified in this work.
Here, we describe the hydrographic data and glider flight data collected in the northern South China Sea during
August 2017. The paper is organized as follows. Section 2 outlines the survey conducted by four gliders. The
estimation of vertical water velocities obtained by gliders is introduced in Section 3, and then the approach for
identifying and classifying the ISWs is described and then validated by a satellite image (Section 4). Finally, the





summary and discussion are presented in Section 5.
**2. Glider observation**
Four Petrel-II gliders (No. Glider-05/06/08/10), as shown in Fig. 1(a), developed by Tianjin University were
prepared for observation of ISWs in the northern South China Sea. According to the statistical analysis of ISWs in
the spatial occurrence from synthetic aperture radar (SAR), the ISWs in the northern South China Sea are mainly
distributed in the region of Luzon Strait, Dongsha Atoll and eastern Hainan Island (Huang et al., 2008). Most ISWs
in the northern South China Sea are generated within Luzon Strait, and propagate westward (Simmons et al., 2011).
Those waves present relatively sparse distribution in the region at over 3000m depth than the vicinity of Dongsha
Atoll. The Petrel-II glider is designed for applications to 1500m depth and perform best for profiles deeper than 600m.
However, in the northwest Dongsha Atoll, the depth of water is less than 500m, and the complicated topography may
endanger the glider. The ISWs are concentrated within a longitudinal band from 117°E to 119°E and a latitudinal
band from 19°N to 22°N. The gliders were deployed in this region where the water depth is over 1000m and run with
approximately parallel trajectories in order to cover this area as wide as possible.
Besides, the occurrence frequencies of ISWs also fluctuate significantly from month to month (Zheng et al.,
2007). The ISWs occur more frequently from April to August. According to marine meteorological conditions, 4
Petrel-II gliders conducted ISWs observation cooperatively in August 2017. Those gliders were deployed in the
northeast of Dongsha Atoll and then proceeded southwestward and back. The trajectories are shown in Fig. 1(b).

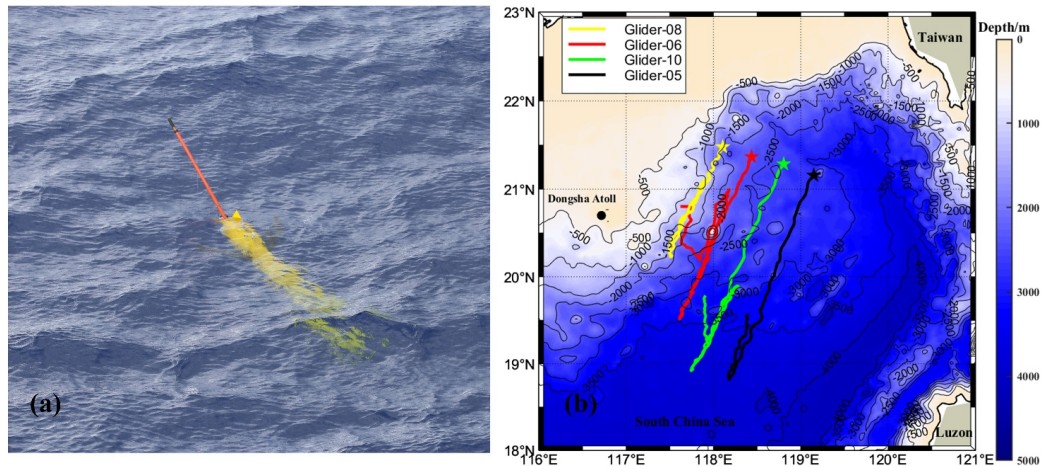


Figure 1(a) A Petrel-II glider at the surface for communication. (b) The trajectories of 4 gliders deployed in the active region of internal
solitary waves, whose starting positions are presented by pentagrams.
During this survey, four gliders yielded 647 profiles and traveled 1907.7km cumulatively (Table 1). In most
profiles, gliders dived into 900~1000m deep, except in a few profiles they did shallow dives to 600m, considering
the steep terrain near the Dongsha Atoll. For the purpose of obtaining high-resolution data, each glider was set to run
with a vertical velocity of 0.1~0.15m/s by adjusting the net buoyancy and pitch angle, and thus, it took 2~5 hours for
the gliders to perform a sawtooth profile with 2~5 km horizontal distance. Each glider is equipped with a Seabird



GPCTD, sampling at 1Hz to record the thermohaline fluctuation with a high resolution when the ISWs pass.
Therefore, the high sampling frequency and low gliding velocity enable the maximum thermohaline resolution to be
0.15m in vertical and 2.5km in spatial scales.

Table 1 Summary of the cooperative observation by several gliders in August, 2017

| No. | Deployment | recovery | profiles | distance (km) |
|---|---|---|---|---|
| Glider-05 | 08.04 | 08.18 | 146 | 397.7 |
| Glider-06 | 08.04 | 08.28 | 195 | 627.6 |
| Glider-08 | 08.05 | 08.21 | 121 | 310.5 |
| Glider-10 | 08.04 | 08.29 | 185 | 571.9 |
| Total | / | / | 647 | 1907.7 |


## 3 Estimation of vertical water velocities

The glider's sawtooth movement in longitudinal plane can be described by the kinematic model (Leonard and
Graver, 2001). Affected by the vertical variations of seawater physical properties and pressure hull deformation, the
net driven force of glider is gradually varied with depth, which causes the glider to perform non-uniform speed
movement during the phase of diving or climbing. However, the acceleration of glider is far below its velocity in
these phases, and acceleration values have a negligible influence on the velocity on a small-time scale. Hence, except
buoyancy changes at the beginning and turning point of profiles at the maximum depth, the stable underwater motion
can be deemed as quasi-steady flight, and the theoretical or still-water vertical velocities ($w_g$) of glider can be
estimated by the kinematic equations, which has been derived in the Ma et al.(2018). The actual vertical velocitiy
($w_p$) calculated from pressure sensor is the sum of the vertical water velocity ($w_c$) and theoretical velocity ($w_g$)
(Merckelbach et al., 2010). Consequently, $w_c$ can be yielded as
$$w_c = w_p - w_g \tag{1}$$

The core of estimating $w_c$ based on the glider lies in the kinematic model. Most of parameters (such as mass,
pitch angle, etc.) in the model are measured through the sensors or tools, but part of parameters, including
hydrodynamics coefficients ($C_{D0}$, $C_D$, $C_{L0}$, $C_L$), coefficient of compressibility ($\gamma$), thermal expansion coefficient ($\varepsilon$),
and glider volume at atmospheric pressure ($V_g$) are indirectly estimated by the empirical formulas or computer
simulations. In addition, gliders work underwater for several days or even months. Therefore, the parameter variations
in the model arising from uncertainties have a remarkable effect on the vertical water velocities. For example,
biofouling can increase the drag force of glider. The model coefficients can be identified and calibrated by minimizing
cost function (Merckelbach et al., 2010; Frajka-Williams et al., 2011; Merckelbach et al., 2019).
Considering the complex calculation and coupled nonlinear coefficients in the model, influence of parameter
variations on the vertical water velocities remains to be determined. A local sensitivity analysis is adopted to explore
the effect of parameter uncertainty on the results, and most obvious factors are optimized by applying a standard
nonlinear search method (Merckelbach et al., 2010).
Gliders work in the time-varying marine environment, so the data acquired in the stable upward motion, where
the roll is zero and the pitch is constant, are input into the glider model, and the sensitivity of each indirectly estimated
parameter which could influence the theoretical vertical velocity is quantificationally presented with single factor





variance analysis. A different boundary condition of each variable is set in numeric computation. During sensitivity
analysis of structural parameters, the glider volume $V_g$ fluctuates $\pm 10$mL around the original value, whereas
coefficient of compressibility ($\gamma$) or thermal expansion coefficient ($\varepsilon$) varies $\pm 10\%$ around the original value. The
effects of those parameters on mean theoretical vertical velocity are calculated separately, and the results are shown
in Fig. 2(a).

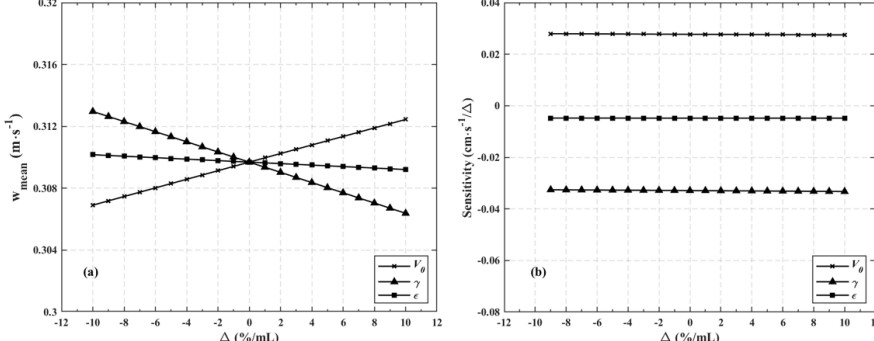


Figure 2 The sensitivity analysis of structural parameters ($V_g$, $\gamma$ $\varepsilon$). (a) The influence of parameter variation on the mean vertical velocity
of model. (b) The sensitivity of parameters in the mean vertical velocity of model.
The results of Fig. 2(a) reveal that the mean theoretical vertical velocity $w_g$ increases approximately linearly
with glider volume $V_g$. As shown in Eq. (5) (Ma et al., 2018), the driven force in the climbing phase augments with
$V_g$, and that results in an increased vertical velocity when the other parameters are constant. By contrast, the mean $w_g$
reduces gradually as the increasing coefficient of compressibility ($\gamma$) or thermal expansion coefficient ($\varepsilon$), which is
caused by the reduction of buoyancy force. The sensitivity of those structural parameters to mean $w_g$ is referred in
Fig. 2(b), and the sensitivity of $V_g$, $\gamma$ and $\varepsilon$ to the mean vertical velocity is 0.0274~0.079 cm•s$^{-1}$/$ml$, -0.032~-0.033cm
•s$^{-1}$/$\Delta$ and -0.004 cm•s$^{-1}$/$\Delta$ ( $\Delta$ denotes a variation rate of 1%), respectively. The sensitivity of $\varepsilon$ is lower than others.
Thereby, the compressibility coefficient ($\gamma$) and glider volume ($V_g$) have a greater influence on the theoretical vertical
velocities.
Similarly, the sensitivity of hydrodynamic coefficients ($C_{D0}$, $C_D$, $C_{L0}$, $C_L$) is analyzed. As is clear in Fig. 3(a),
the mean $w_g$ reduces gradually as the increase of $C_{D0}$ or $C_L$, while the mean $w_g$ changes little along with the increasing
$C_D$ or $C_{L0}$. The sensitivity of mean $w_g$ to those hydrodynamic coefficients is given in Fig. 3(b). The sensitivity of $C_{D0}$
and $C_L$ is -0.11~-0.07 cm•s$^{-1}$/$\Delta$ and -0.068~-0.047 cm•s$^{-1}$/$\Delta$, respectively, while the sensitivity of $C_D$ or $C_{L0}$ is close
to zero.



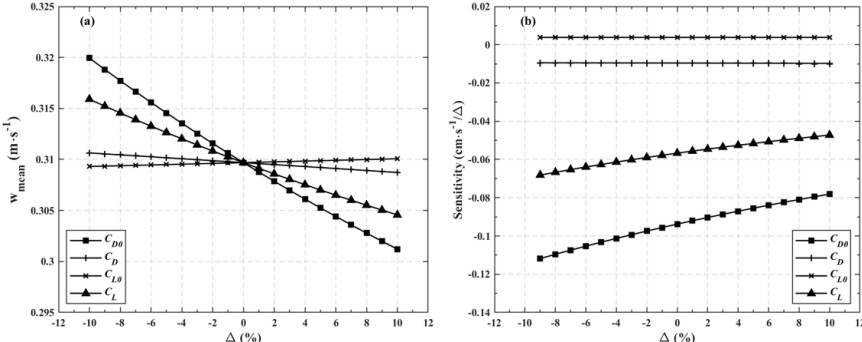


Figure 3 The sensitivity analysis of hydrodynamic coefficients ($C_{D0}$, $C_D$, $C_{L0}$ and $C_L$). (a)The influence of coefficient variations on the
mean vertical velocity of model. (b) The sensitivity of coefficients to the mean vertical velocity of model.
According to the sensitivity analysis results, structural parameters ($V_g$, $\gamma$) and hydrodynamic coefficients ($C_{D0}$,
$C_L$) own a high sensitivity to the theoretical vertical velocities ($w_g$), while others cause an influence on $w_g$ with the
order of $O(10^{-3})$, which can be ignored. Therefore, those high-sensitivity factors are ameliorated by minimizing the
cost function $F$, thus to reduce the error of estimated vertical water velocities (Merckelbach et al., 2010).
Given the variable running depth and unstable motion in the phase of pumping oil, the data obtained at depths
shallower than 600 m are utilized for the minimization process. Those optimized parameters as a function of dive
number are shown in Fig. 4.

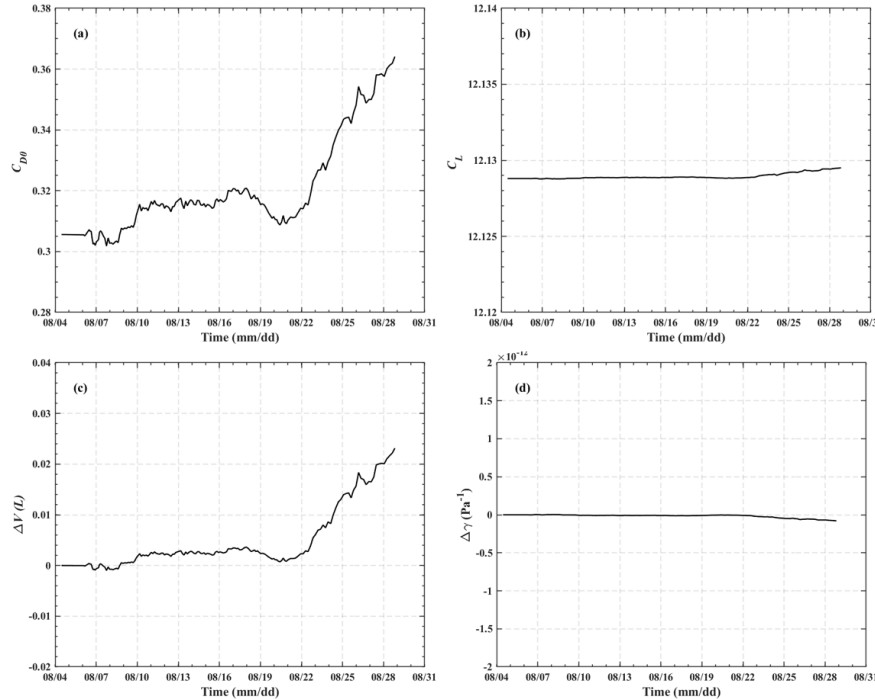


Figure 4 Optimized parameters versus time. (a) drag coefficient. (b) lift coefficient. (c) glider volume. (d) compressibility coefficient.


The optimization results show that the drag coefficient $C_{D0}$ increases with working time. The biofouling by
barnacles and film-like sludge is the main reason that causes drag increase. Those attachments are clearly visible on
the recovered glider (Fig. 5). However, the lift coefficient $C_{L0}$ has no significant difference relative to $C_{D0}$. The
buoyancy change generates driving force for glider's sawtooth motion. When the glider dives into the target depth,
the hydraulic oil are pumped into the external bladder from the inner tank inside the pressure hull. Under the influence
of repeated cold-heat cycles, the air bubbles dissolved in the oil are separated, and occupy a small proportion of the
inner tank volume, which gradually increase with the number of profiles. Petrel-II glider adjusts its net buoyancy
according to the detected volume change of the inner tank. The air bubbles lead to the fluctuation of glider volume
$V_g$, and still exist in the inner tank when we maintain the glider after recovery. Those air bubbles may influence the
compressibility $\gamma$ of glider.

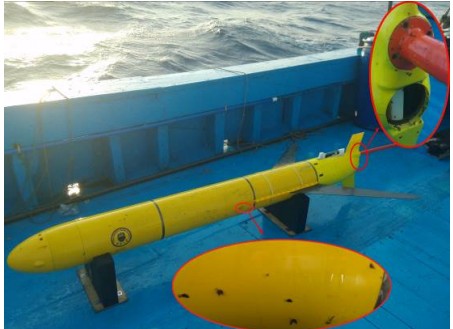


Figure 5 The biofouling on the glider hull. The surface of glider is attached by the barnacles and film-like sludge.
The structural parameters can be validated through the depth-keeping experiment. The glider's buoyancy should
be equivalent to its gravity to keep the glider in the constant depth. Hence, this motion can be realized by setting
appropriate buoyancy. The depth-keeping experiment and simulations with original and corrected structural
parameters are shown in Fig. 6. The experiment lasted for about half an hour, and the glider held depth at 848±1m.
Substituting the original and corrected parameters $V_g$ and $\gamma$ separately into the buoyancy model yields the depth
simulations, and the error of depth decreases from 24m to 12m when corrected parameters are adopted.

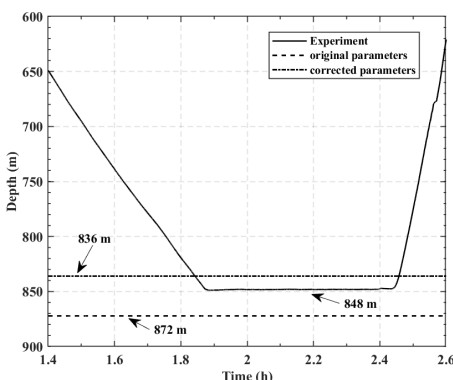


Figure 6 Comparison between simulation and depth-keeping experiment. The error is reduced by half with the adoption of corrected
parameters.





The glider vertical velocity $w_p$ relative to water velocity is obtained by the time rate of change of pressure
measured by the CTD, but those signals contain noises, or even glitches. Due to the excellent time-frequency
characteristic, the wavelet transform is applied to restrain those noises. Based on the vertical velocity estimation
method and optimized parameters, the water and glider vertical velocities are achieved, as shown in Fig. 7.

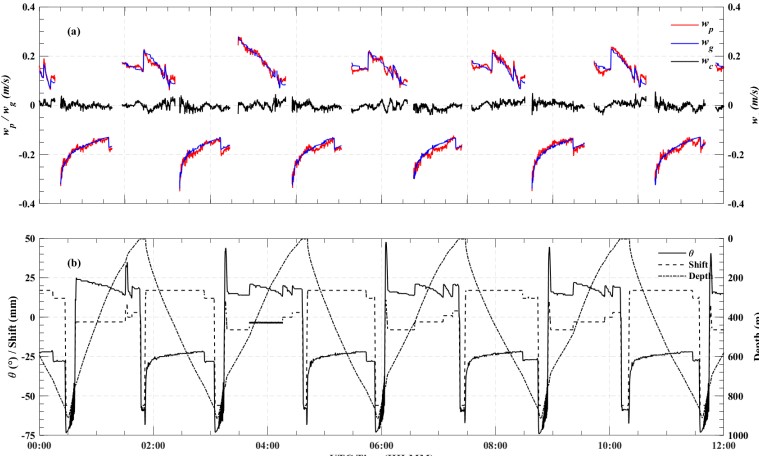


Figure 7 Vertical velocity, pitch, shift, depth versus time. Given the unsteady motion during the eccentric battery pack shifting or rotating,
and variable buoyancy engine working, the velocities in those moment are excluded.
The tendency of theoretical velocity ($w_g$) is coincident with that of vertical velocity ($w_p$) in Fig. 7. The error of
the estimation of vertical water velocity inferred from glider data is mainly due to the mixed sampled noise and the
inaccuracy of parameters in the model. It is difficult to compare the estimated results with the vertical velocities
measured independently in the field. A robust approach to estimate the vertical water velocities is proposed in the
reference (Merckelbach et al., 2010). The mean vertical water velocities for 3-day periods are 2.22±0.41mm/s after
parameter optimization, and the fluctuation of mean values in the adjacent bins of 50m is below 0.1mm/s, showing
that values share well continuity in the vertical direction. Over the same time periods, the offset of vertical water
velocity between dives and climbs is -2.9±1mm/s. Therefore, the inaccuracy of vertical water velocity estimated from
glider data is nearly 4mm/s.

## 204    4 Identifying internal solitary waves

Four gliders conducted the observation cooperatively in the mission, moved southwestward and then traveled
backward. The vertical water velocities are calculated with the method described in Section 3. We analyze the vertical
water velocities within the common depth from 50m to 500m during the steady gliding motion. Therefore, the spatial
distribution of the standard deviation (*std*) of $w_c$ during each dive in this region from 4 August to 16 August is mapped
out as shown in Fig. 8. The pink chain-dotted lines denote the observing time at intervals of 1 day.



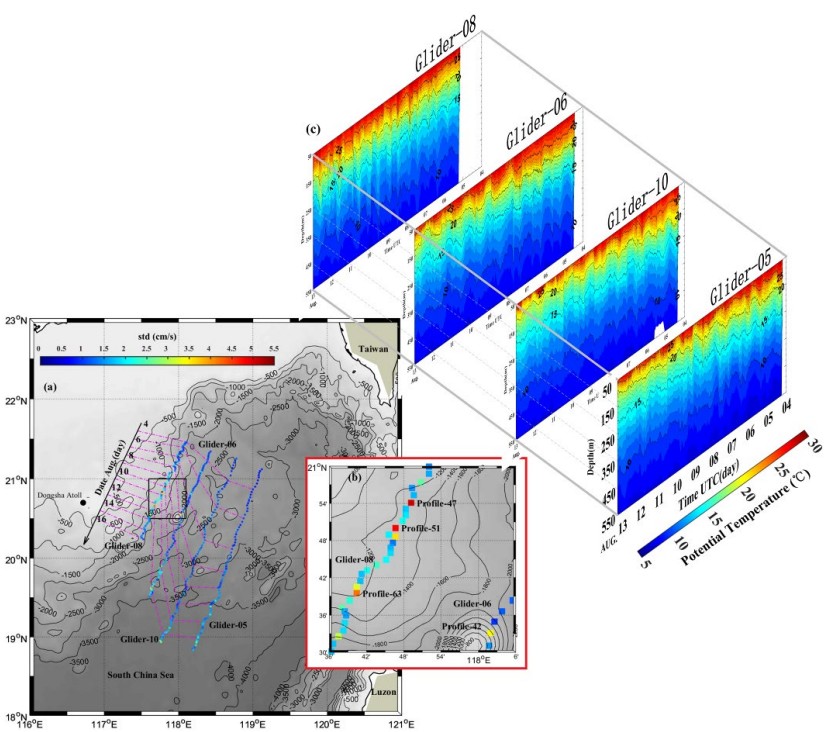

Figure 8 (a) Spatial distribution of the standard deviation (std) of $w_c$, (b) the dramatic *std* in the observation region of Glider-08, and (c)
potential temperature obtained by glider-based GPCTD between 4 August and 12 August (50m~500m). Color shadings and contour lines
are isothermals at 1℃ intervals. Isothermal surfaces of 10, 15, 20, and 25℃ are labeled.
The *std* of inferred vertical water velocities during each glider dive exhibits a significant variation in the
coverage area of observation, and can be denoted the turbulent mixing or the internal wave strength (Beaird et al.,
2012; Todd, 2017; Evans et al., 2018). This variation indirectly reflects that the activity of vertical water increases
gradually at the same latitude along the east-west direction, and such distribution coincides with fluctuation of
isotherms in the 4 temperature transections measured by glider-based CTD. This phenomenon may be due to the
combination of the nonlinear effects and topographic effects when the waves are propagating westward (Simmons et
al., 2011).
The average *std* of all inferred vertical velocity profiles is 1.5±0.5 cm/s within that depth range. Glider-08
encounters violent vertical flow when passing the rough topography in the survey, which may be caused by the
passage of ISWs on the east side of Dongsha Atoll, shown in Fig. 8(b). The standard deviations of vertical velocities
present a dramatic rise up to 5.06cm/s (20°54.08′N, 117°49.23′E), 4.83cm/s (20°50.02′N, 117°46.67′E), and 4.22cm/s
(20°39.48′N, 117°40.42′E), respectively, and these values are considerably larger than the average.
Waves with various structures, namely depression wave and elevation wave, can cause disparate fluctuation of
thermohaline structures. The passing depression wave forces isotherms to move sharply down, while the passing
elevation wave induces an opposite displacement (Fu et al., 2012). In addition to thermohaline fluctuations, another
notable phenomenon induced by ISWs is the sharp horizontal water velocity, and those ISWs in the northeastern SCS
propagate westward at a speed in the order of 0.72 to 1.8 m/s (Liu et al., 2004). With the aid of hydrodynamic force





generated by wings, the horizontal velocity of glider is typically about 0.2~0.4m/s. The horizontal speed gap between
glider and ISW can affect the way of encountering, so generally the ISW propagates past the glider. In other words,
if an ISW passes the present position of glider, the glider cannot capture this wave again whichever direction the
glider travels along, even the glider travels along the direction of the wave propagation.
Generally, the magnitude of vertical water velocities induced by ISWs is larger than that of the background
velocities before arrival of ISWs. The occurrence of ISWs can be reflected by the vertical water velocities estimated
by the glider (Rudnick et al., 2013). This, together with characteristic of ISWs, can be used to further analyze those
extraordinary profiles in Fig. 8(b). The time series of vertical water velocities and depth of three continuous profiles
(No.46/47/48) performed by Glider 08 are shown in Fig. 9.

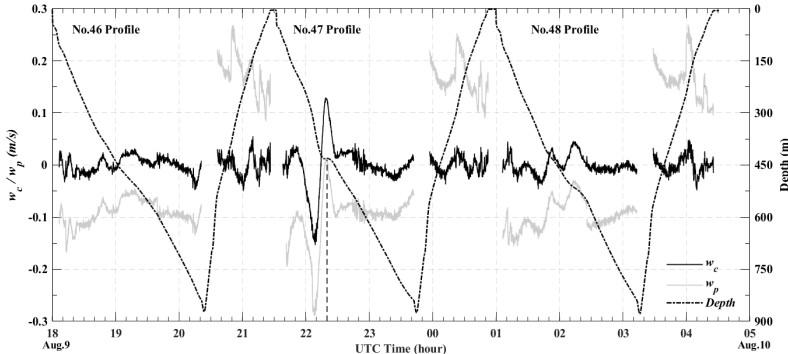


Figure 9 The vertical velocities derived from Glider08 data and depth as a function of time
As obvious from Fig. 9, the vertical water velocity derived from Profile-47 experiences a dramatic fluctuation
at the depth 150-450m in the diving phase. As the depth increases, the vertical currents present large downward
velocities, which then convert into upward velocities in less than 40 min, and the peak magnitudes of upwelling and
downwelling are about 0.13 m/s and 0.14 m/s, respectively. The strong upwelling forces the glider to change its
predefined movement direction from downward to upward at 22:20 UTC. After that, the vertical currents present a
relatively gradual change, and in the adjacent profiles, no significant perturbations appear in the vertical dimension.
Taking the chronological change of vertical waver velocities into consideration, this rapid phenomenon occurring
during the diving process of Profile-47 is consistent with the passage of the abrupt ISW. Only a complete peak-to-
trough vertical velocity oscillation over such a period of time suggests that the wave is likely a single-wave ISW.
The glider can synchronously acquire thermohaline structure with the payload CTD, and the thermal
stratification is shown in Fig. 10. The passage of ISWs captured by Profile-47 induced sunken displacements of
isothermals, and this phenomenon coincides with characteristics of a depression ISW.

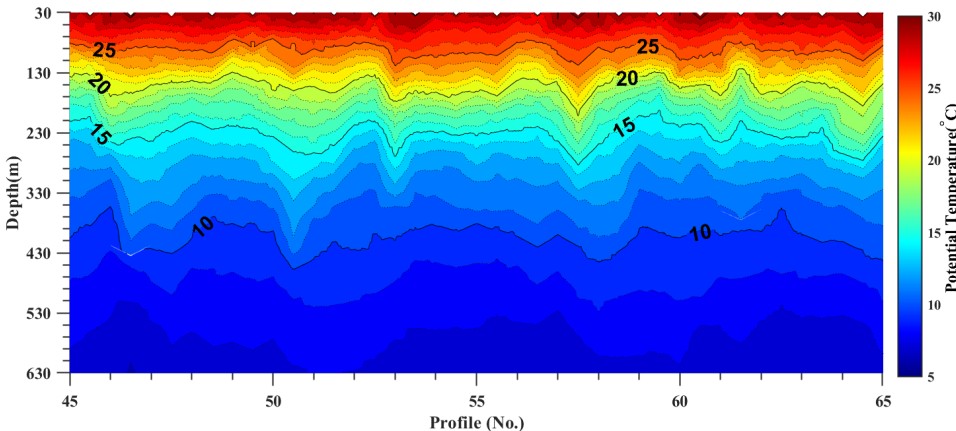


Figure 10 Thermal structural distribution obtained by Profiles 45~65 (Glider-08)

Another interesting phenomenon occurs in profiles 61~65. As is clear in Fig. 10, it seems that a small depression
wave arrived in profile-61, making the thermocline sunk to nearly 300m depth. Then the thermocline quickly
recovered in the following profiles 62~65 suggesting that the wave influencing profile-61 had already passed. After
that, the isothermals at water depths from 100m to 400m fluctuate dramatically again in profile-65, which may be
impacted by another depression wave.
The vertical water velocities derived from those profiles of the glider are taken for further analysis. In clear
contrast to relatively stable thermohaline structure in profiles-62~65, the vertical water velocities of those profiles
exhibit consecutively periodic oscillations as illustrated in Fig. 11. The vertical velocities of the wave completing a
cycle of peak- trough- peak oscillate in the form of a simple sinusoidal independent of time (Todd, 2017). Given that
the horizontal velocity of glider is far smaller than the propagation speed of ISW, the glider cannot cross the same
ISW again. Those consecutively sinusoidal oscillations of vertical water velocities inferred from Profile-63 are likely
influenced by a multiple-wave packet with a train of rank-ordered ISWs.

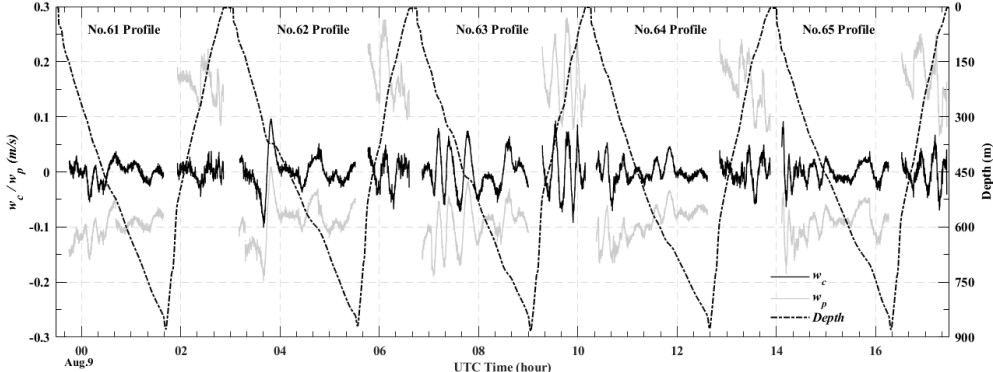



Figure 11 Vertical velocity, depth versus time of Profiles 61~65 (Glider-08)

The ISWs cause the change of roughness on the sea surface, and alter the sun-glint reflection. These
characteristics are presented with the bright or dark strips in the satellite images, which is often used to catalog the



occurrence of ISWs (Zhao et al., 2004). One true-color MODIS (Moderate Resolution Imaging Spectroradiometer)
satellite image (Fig. 12) with 250m-resolution taken on 12 August 2017 at 3:15 UTC presents a snapshot of the active
internal waves in the South China Sea. The image displayed in a partially enlarged frame clearly shows the existence
of two types of waves, a single-wave ISW (left arrow pointing) and a multiple-wave packet (right arrow pointing).
Coincidentally, the Glider-08 conducting the No.62 profile (Fig. 11) is located at the western edge of the multiple-
wave packet at the same time. Since the waves propagate westward, there is a high possibility that glider-08 captures
the multiple-wave packet in the continuous profiles, and the probability is verified by the oscillation of vertical water
velocities inferred from glider.

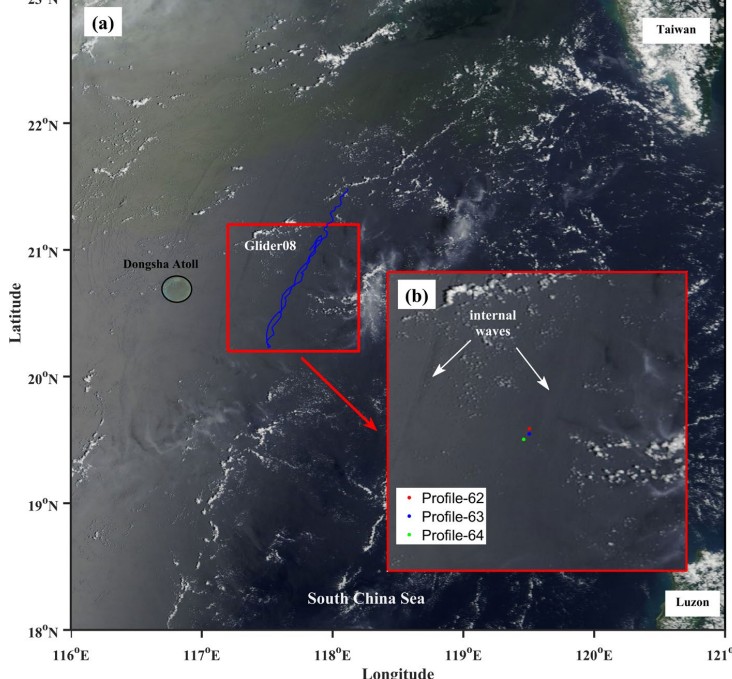


Figure 12 A MODIS true-color image of ISWs (resolution: 250m) in the South China Sea acquired at 03:10 UTC, Aug.12, 2017. The
blue line in (a) is the Glider 08's trajectory. The red solid box is enlarged in (b), and the red, blue and green points denote the valid GPS
position of No.62, 63 and 64 profiles, respectively.

The thermohaline structure obtained by glider-based CTD reflects less features of ISWs than the vertical water

velocities inferred from glider data. This is due to the movement characteristics of glider. The sawtooth motion of
glider makes it unable to suspend at an appointed position for a long time, and thus the glider cannot carry out the
measurement of the vertical seawater properties at multiple depths synchronously, which is adopt by the sensor chains
in the mooring. Considering the significant speed differences between glider and ISW, it is so hard to determine
whether the thermohaline perturbation observed by glider-based CTD is induced by a single-wave or multiple-wave
packet, and even whether there exist waves within the spurious stable thermohaline stratification inferred from glider-
based CTD data.

The vertical water velocities deduced by the glider provide a robust evidence that ISWs have passed by the



glider's current position. The opposite vertical velocities are generated on the leading and trailing edges of wave
along its propagation direction (Ramp et al., 2004). Ideally, the oscillation of vertical water velocities presents a
sinusoidal form when the glider passes successively through the leading and trailing edges of a wave, and this
phenomenon occurs continuously if the glider flies through a multiple-wave ISW packet. Hence, it is more
intuitionistic and accurate to determine the existence and type of the ISW by analyzing the feature of vertical water
velocities inferred from glider, and this method is tested to be practicable. Further, in combination with the convex
or concave isotherms/isohalines obtained by glider-based CTD, the characteristic (depression or elevation) of wave
can be determined.

## 5. Summary and discussion

We applied several underwater gliders to observe internal waves in the South China Sea. Without extra current
sensors, the vertical water velocity is derived through the combination of the quasi-steady flight model and sea-trial
data of glider. The accuracy of parameters in the model directly determines the credibility of the inferred velocity.
Therefore, the local sensitivity analysis method is applied to discuss the parameters in the model, which cannot be
measured exactly by the sensors or tools. The results indicate that structural parameters ($V_g$, $\gamma$) and hydrodynamic
coefficients ($C_{D0}$, $C_L$) are the main factors affecting the accuracy of the inferred vertical water velocity. Those
predominant parameters are calibrated by nonlinear optimization algorithm. Furthermore, the hover experiment
validates the effect of the optimized structural parameters, and the error of depth range is reduced by half of the
amount. With the same error estimation method of Slocum gliders (Merckelbach et al., 2010), the accuracy of the
inferred vertical water velocity is nearly 4mm/s.
The observation with 4 gliders is conducted in the SCS. The *std* of the inferred vertical water velocity during
each dive characterizes the strength of vertical water activity, and exhibits a gradual increase in the same latitude
along the east-west direction in the coverage area of observation. This phenomenon is coincident with the fluctuation
of isotherms.
A few profiles, where the *std* of vertical water velocity is larger than the average *std,* are further analyzed. Since
those gliders are deployed in the region between Dongsha Atoll and Luzon Strait, where ISWs occur with high-
frequency in July and August (Zheng et al., 2007), the dramatic vertical water mass flow may be attributed to the
propagation of ISWs. The glider traveling along with the sawtooth motion cannot maintain a fixed position like
moorings to obtain complete thermohaline stratification at different depths before and after ISWs arrival. Therefore,
the common identifying and classifying method of ISWs using the temperature structure measured by glider CTD
may miss the key feature of ISWs.
Given vertical flow induced by the passage of ISWs, the vertical water velocity inferred from glider, together
with the thermohaline perturbation, is utilized to identify the ISWs. The vertical velocities of the wave in a complete
cycle peak- trough- peak experience an approximate sinusoidal oscillation. The horizontal speed gap between the
glider flying and ISW propagation determines that the glider cannot travel across the same ISW repeatedly. Therefore,
analysis of the vertical water velocity oscillation in the velocity profile with remarkable *std* and adjacent profiles
helps to determine whether the encountered wave is a single- or multiple-wave ISW packet. Glider-08 captures
different kinds of ISWs in the sea-trial, and the method is proven to be feasible and effective by comparison with a



MODIS image. Furthermore, according to the sunken isothermals, those captured waves are classified as depression
waves.
This paper investigates the activity of internal waves in the eastern area of Dongsha Atoll and proposes a method
of identifying the type of ISWs, which is applicable to glider observation. Although gliders may enable us to obtain
high-resolution observation data, there are comparable challenges to estimate the key parameters of ISWs, such as
propagation direction and the phase velocities. Direct velocity measurements using current meters or current profilers
on gliders might provide a solution to this problem. Future cooperative surveys with a fleet of gliders can be
performed to understand the propagation and evolution of ISWs.
*Data availability. T*he MODIS image is available at https://earthdata.nasa.gov/earth-observation-data. Data from field
experiments is available on request from wei.ma@tju.edu.cn or corresponding author.
*Author contributions*. Wei Ma carried out the research and initiated the paper. All the authors collected, processed
and analyzed the observations and contributed to revisions and comments on the paper
Competing interests. The *authors* declare that they have no conflict of interest.

*Acknowledgements.* This work is supported by the National Natural Science Foundation of China (Grant
Nos.52005365) and National Key R&D Program of China (Grant Nos. 2016YFC0301100).

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
