# Peer review of "The inference of internal solitary waves in the northern South China Sea from"

_Ocean Science, 2021_

## Author Comment (AC1)

**Comment on os-2021-29**

Anonymous Referee #1

Dear Referee #1:

We have carefully studied the comments of all reviewers and have improved the manuscript accordingly. We appreciate invaluable constructive comments of the reviewers, which we believe have substantially strengthened the manuscript. In this letter the comments are discussed in detail and the modifications are highlighted in blue in the revised manuscript.

Once again, we appreciate yours and that of the reviewers' time and effort, and hope that we have addressed the comments adequately. We remain available for any further modifications and look forward to hearing from you.

Sincerely yours,

Wei Ma

On behalf of all authors.

**General comments**

**Comment:** The authors present a method of detecting Internal Solitary Waves (ISWs) using unmanned underwater gliders. The glider flight data and hydrodynamic model are used to estimate the vertical velocity of the water, which is used, in conjunction with the temperature data recorded by the glider, to detect the presence of ISWs. This is confirmed by comparison with a coincident MODIS satellite image.

In principle, the method sounds promising. Unfortunately, the description of the method is inadequate. It would be impossible for anyone else to reproduce this method from the information presented here, and it is not clear how their method for estimating vertical water velocities from gliders differs from previous authors.

There is also a noticeable lack of discussion. No comparisons are made with previous studies of ISWs, or with previous studies detecting internal waves from gliders. There is no discussion of the wider applicability of their method outside the South China Sea (where ISWs are known to occur with very large amplitudes). There is virtually no discussion of the advantages of using gliders for this work over other measurement platforms apart from some extremely general remarks in the introduction. (For example, in the Introduction it is mentioned that surveys from research vessels are time consuming and costly, but there is no discussion later on of the scientific advantages or disadvantages of using gliders to observe ISWs instead of research vessel surveys.)

I also do not feel this article is a good fit for Ocean Science. Research articles are supposed to report "substantial and original scientific results", which this article does not do since it is almost entirely about the method. It could possibly be considered for a technical note but is perhaps too long. Perhaps a journal more focused on methods and technology, such as the Journal of Atmospheric and Oceanic Technology, would be a better match?

**Response:** We would like to thank the reviewer's great efforts in reading our manuscript and for your constructive comments and suggestions. Our responses to the comments and suggestions are listed as follows. We would like to know if there are still somewhere need to be amended.

The following changes were made :

[Lines 369-387] "This paper investigates the activity of internal waves in the eastern area of Dongsha Atoll. The approach and accuracy to estimation of vertical water velocity reported by previous authors (Merckelbach et al., 2010; Frajka-Williams et al., 2011; Rudnick et al., 2013) is not be discussed in detail herein, because those methods have been proven effective and applied in the research of internal waves. We conducted the exploratory work of addressing the question of whether gliders can detect ISWs and identifying the type of ISWs, which is applicable to glider observation, and have been demonstrated around the Dongsha Atoll in the SCS. In the further, the applicability of this method will be verified by experiments in regions where the ISW amplitudes are smaller than in the SCS.

This work is an interesting trial of studying the ISWs with glider observation. Unlike mooring array, gliders can adjust the observation path dynamically. Furthermore, their yo-yo motion allows for sampling of subsurface features of ISWs which cannot be obtained by satellites. The observation capabilities of glider are similar to the shipboard lowered CTD, but the long-term and continuous observation for obtaining high resolution time series of ISWs poses a challenge to the latter. The glider with characteristic of slow profiling speed is suitable for observing the high-frequency internal waves (Rudnick et al., 2013). However, it is a disadvantage that gliders observe temporal evolution of internal waves considering the weak hold-station capability in the strong current region (Todd, 2017). Besides, although gliders may enable us to obtain high-resolution observation data, there are comparable challenges to estimate the key parameters of ISWs, such as propagation direction and the phase velocities. Direct velocity measurements using current meters or current profilers on gliders might provide a solution to this problem. Future cooperative surveys with a fleet of gliders and multiple observational platforms can be performed to understand the propagation and evolution of ISWs better."

This work is a first step on studying the ISWs with glider observation. We focus on addressing the question of whether gliders can detect ISWs and how to identify the type of ISWs based on the glider data. However, to our knowledge, those works have not been reported in the literature. We only conducted the ISW observation in the SCS. It is a pity that Petrel-II glider have not been used outside the South China Sea to verify this method. We hope to have the opportunity to carry out this experiment in the future.

**Major concerns**
**Section 3: Estimation of vertical water velocities**

**Comment:** Correctly estimating the vertical velocity of the water is vital for this work. Unfortunately, section 3 is not well written, to the extent that I am unable to determine if the results presented here are valid or not. The main problem is that the authors have simply left out significant parts of their methods. It is always difficult, when building on the work of previous authors, to strike a balance between not repeating too much of what has been published previously and yet still creating an article which is reasonably comprehensible on its own. However, the authors have gone too far towards not repeating material published previously. I suggest the authors compare with the article by Fer et al., 2014, JAOT. Their section 4(a) describes the glider flight model, including the basic kinematic equations and definitions of every parameter. Fer et al. explain which parameters are simply given optimized values from previous published work, which are calculated from the glider's data, and which parameters are optimized for their dataset and which cost function is used to do so (this is very important, there are several possible cost functions). This whole section is obviously very heavily

based on the previously published work of Merckelbach et al., 2010, but Fer et al. strike a good balance between not repeating too much of Merckelbach's work while still including enough that the method is comprehensible without reading Merckelbach's paper. I suggest the current authors rewrite the first part of their section 3 along the same lines.

The next part of section 3 concerns the sensitivity analysis of the various flight model parameters. However, the authors do not explain where the 'original values' of each parameter come from, nor do they explain how the effects of these parameters on the mean theoretical vertical velocity are calculated, or how they calculate the local sensitivities of these parameters. Once again it is impossible to follow their method. On line 142 they refer to an equation 5 when only one equation has been presented at this point in the paper (possibly this is meant to refer to Equation 5 in a 2018 paper by Ma et al., but if the equation is important enough that it is necessary to refer to it specifically, then it's important enough to be reproduced in the current paper).

**Response:** Thank you very much for your comment. The following changes were made:

[Lines 113-143]: "The glider's sawtooth movement in longitudinal plane can be described by the kinematic model (Leonard and Graver, 2001). Affected by the vertical variations of seawater physical properties and pressure hull deformation, the net driven force of glider is gradually varied with depth, which causes the glider to perform non-uniform speed movement during the phase of descent or ascent . However, the acceleration of glider is far below its velocity in these phases, and acceleration values have a negligible influence on the velocity on a small-time scale. Hence, except buoyancy changes at the beginning and turning point of profiles at the maximum depth, the stable underwater motion can be deemed as quasi-steady flight, and the balance of horizontal vertical forces applied to the glider in this flight mode can be expressed as:

$$\begin{cases} 0.5\rho(C_{L0}+C_L\alpha)A_DV^2\sin(\theta+\alpha)-0.5\rho(C_{D0}+C_D\alpha^2)A_DV^2\cos(\theta+\alpha)=0 \\ F_g-F_b-0.5\rho(C_{D0}+C_D\alpha^2)A_DV^2\sin(\theta+\alpha)-0.5\rho(C_{L0}+C_L\alpha)A_DV^2\cos(\theta+\alpha)=0 \\ F_b=\rho g\left\{V_g[1-\gamma P+\varepsilon(T-T_0)]+\Delta V_b\right\} \\ F_g=m_{total}g \end{cases} \tag{1}$$

where $F_g$ is the glider's weight, $F_b$ is the buoyancy force, $\rho$ is density *in situ*, $A_D$ is the maximum cross-sectional area of the glider, and $V$ is the gliding velocity through water along the glide path, $\theta$ is the pitch angle, and $\alpha$ is the attack angle, which is the numerical solution by solving the first expression in the Eq.(1); $C_{L0}$, $C_L$, $C_{D0}$ and $C_D$ are the coefficients of lift and drag; $V_g$ is the glider volume at atmospheric pressure, $\gamma$ is coefficient of compressibility, $P$ is the water pressure, $\varepsilon$ is the thermal expansion coefficient, $T$ is the water temperature, $T_0$ is a reference water temperature, and $\Delta V_b$ is buoyancy change provided by the external bladder; $m_{total}$ is the total mass of glider, and $g$ is the gravitation acceleration *in situ*. The deduction process in detail has been reported by Ma et al. (2018).

The theoretical or still-water vertical velocities ($w_g$) of glider can be inferred as

$$w_g = V\sin(\theta+\alpha) \tag{2}$$

The actual vertical velocity ($w_p$) calculated from pressure sensor is the sum of the vertical water velocity ($w_c$) and theoretical velocity ($w_g$) (Merckelbach et al., 2010). Consequently, $w_c$ can be yielded as

$$w_c = w_p - w_g \tag{3}$$

The core of estimating $w_c$ based on the glider lies in the parameters of kinematic model. In Eq.(1), $\theta$ in the model are measured through the digital compass; $m_{total}$ ~73.7 kg; $A_D$=0.0484 m²; and $\rho$ is calculated based on

the conductivity, $T$ and $P$ measured by GPCTD. However, part of parameters, including hydrodynamics coefficients ($C_{D0}$, $C_D$, $C_{L0}$, $C_L$), $\gamma$, $\varepsilon$, and $V_g$ are indirectly estimated by the empirical formulas or computer simulations, and the original values of those parameters are reported in Ma et al., (2018). Note that an approximate up-down symmetrical shape of Petrel-II is employed, so the hydrodynamics coefficients are taken to be same for ascents and descents. In addition, gliders work underwater for several days or even months. Therefore, the parameter variations in the model arising from uncertainties have a remarkable effect on the vertical water velocities. For example, biofouling can increase the drag force of glider. The model coefficients can be identified and calibrated by minimizing cost function (Merckelbach et al., 2010; Frajka-Williams et al., 2011; Merckelbach et al., 2019)."

**Comment:** Importantly, it is not clear what *use* the authors make of this sensitivity analysis, other than to decide which flight model parameters need to be optimised. The list of parameters they decide need to be optimised are the volume, lift and drag, and compressibility. The first three are commonly optimised by glider users, and the last is obviously going to be significant in gliders with air bubbles, such as the authors mention on lines 172-175. Hence it is not clear how their method for estimating vertical water velocities from gliders differs from previous authors.

**Response:** The accuracy of parameters in the flight model is vital for estimating the vertical water velocity. We adopt the different expression of hydrodynamic force, which differs from previous authors. [Line 144-147] "Considering the complex calculation and coupled nonlinear coefficients in the model, influence of parameter variations on the vertical water velocities remains to be determined. A local sensitivity analysis is adopted to explore the effect of parameter uncertainty on the results. The most obvious factors are optimized by applying a nonlinear least squares method (Merckelbach et al., 2010; Fer et al., 2014)."

**Comment:** Lines 130-131 say "the data acquired in the stable upward motion, where the roll is zero and the pitch is constant, are input into the glider model". Does this mean that only data from glider ascents were used to tune the flight model parameters, not descents? What if the drag coefficients were not the same for ascents and descents? Is it a feature of Petrel- II gliders that they do not roll or pitch during ascents? Or something specifically set by the pilots? This needs explanation because other types of glider do roll and make small pitch adjustments during ascents. And if Petrel-II gliders do not pitch during ascents, what effect does this have on the flight (particularly the angle of attack) when encountering changes in water density?

**Response:** Petrel- II gliders roll or pitch during ascents and descents. The data from glider ascents and descents were used to tune the flight model parameters.

[Line 153-156] "In the sensitivity analysis process, considering that gliders work in the time-varying marine environment, we chose the data acquired in the stable upward motion during a profile, where the roll is zero, the pitch is constant and buoyancy engine is deactivated, and different value of structural parameters are input into the glider model to yield $w_g$."

[Line 180-183] "Given the variable running depth and unstable motion in the phase of pumping oil, the data obtained at those time are not utilized for the minimization procedure. Here, the available data points in 3-day-long sets of descents and ascents are used in the minimization process, and a nonlinear least squares method is applied to search the optimal solution of those high-sensitivity factors."

**Comment:** Lines 188-190: "The glider vertical velocity $w_p$ relative to water velocity is obtained by the time rate of change of pressure measured by the CTD, but those signals contain noises, or even glitches. Due to the excellent time-frequency characteristic, the wavelet transform is applied to restrain those noises." This requires a great deal more explanation. It is not common for glider pressure sensors to have glitches. What type of glitches and how severe were they? How did the authors determine that these were glitches? What wavelet transformation is applied, exactly, and what effect does it have on the calculated values of $w_p$? This could be crucial for the final values of $w_c$.

Towards the end of section 3, the authors give mean vertical water velocities for three-day periods of 2.22mm/s, but do not feel the need to discuss these unusually large values.

Most estimates of vertical water velocity suggest values around $10^{-5}$ m/s in the upper ocean (e.g., Liang et al., 2017, JGR Oceans). It is not stated whether these large velocities were upwelling or downwelling, nor is there any discussion of why this region has such large values. I did find myself wondering if these were speeds rather than velocities, which would still be very large but not quite so unusual.

The authors also seem unconcerned that the estimates of vertical water velocities from descents and ascents differ by 2.9mm/s on average. Again, this difference is two orders of magnitude greater than most estimates of vertical water velocity. This raises questions about the effectiveness of their cost function, and/or whether they should use a different cost function to minimize differences between descents and ascents. But since the authors do not state which cost function, they use it is hard to understand how such a large difference was considered acceptable. I feel the authors need to devote some effort towards convincing the reader that their results are still valid even though the difference between vertical water velocities estimated from dives and from climbs is two orders of magnitude greater than previous estimates of vertical water velocity.

**Response:** The measured vertical velocity wp inherits the high frequency noise from the pressure transducer signal (Merckelbach et al., 2010). The accuracy and valid wp is crucial for the estimation of vertical water velocity.

[Line 212-216]: "The actual vertical velocity $w_p$ is obtained by the time rate of change of pressure measured by the CTD. The pressure signals contain noises, or even glitches because of the high frequency sampling of CTD and data storage with the accuracy of one-hundredth of a meter. The application of 3-sigma rule in determinant the glitches of measuring results are distinguished and removed. Due to the excellent time-frequency characteristic, the wavelet signal denoising is applied to restrain those noises."

These large velocities were upwelling. The positive value of the vertical velocity means upward direction.

[Line 229-234]: "A robust approach to estimate the error of vertical water velocities is proposed in the reference (Merckelbach et al., 2010). The mean vertical water velocities for 3-day periods are $2.22 \pm 0.41$ mm s$^{-1}$ (mean $\pm$ standard deviation) after parameter optimization, and the fluctuation of mean values in the adjacent bins of 50 m is below 0.1 mm s$^{-1}$, showing that values share well continuity in the vertical direction. Over the same time periods, the difference between mean profiles of vertical water velocity from ascents and descents is $-2.9\pm1$ mm s$^{-1}$."

Most estimates of vertical water velocity suggest values around $10^{-5}$ m/s in the upper ocean. But given the active internal waves in the observation region of gliders, these unusually large values are possible. The convection events with strong vertical water motion, up to the order of 10 cm s$^{-1}$ had been reported by Marshall and Schott (1999). Frajka-Williams used the method as detailed in the Merckelbach et al. (2010) to evaluate the accuracy of the vertical water velocity from seaglider. "Over the same 3-day periods, we compare the vertical velocity from dives with climbs. The offset of $w_d$- $w_c$ is $-3\pm2$ and $2\pm5$ mm s$^{-1}$ for sg014 and sg015,

respectively. Over the same 50-m depth bins, the offset is 0.007 $\pm$0.9 and 22.3 $\pm$0.9 mm s$^{-1}$ for noise levels in the spectrum. " (Frajka-Williams et al., 2011).

Marshall, J., Schott, F., 1999. Open-Ocean Convection: Observations, Theory, and Models. Reviews of Geophysics. 37(1), 1-64.

Merckelbach, L., Smeed, D., Griffiths, G., 2010. Vertical Water Velocities From Underwater Gliders. J. Atmos. Oceanic Technol. 27(3), 547-563.

Frajka-Williams, E., Eriksen, C.C., Rhines, P.B., Harcourt, R.R., 2011. Determining Vertical Water Velocities From Seaglider. J. Atmos. Oceanic Technol. 28(12), 1641-1656.

**Section 4**

**Comment:** Under the assumption that the authors' methods are valid (despite the lack of clarity in section 3), section 4 presents some fairly convincing results that the authors have successfully detected ISWs using gliders. However, this was in in a region where ISWs are known to occur and are known to have extremely large amplitudes. What about in the rest of the ocean? The introduction begins by saying that ISWs can propagate over thousands of kilometres. I don't think the authors have demonstrated the wider applicability of this method in regions where the ISW amplitudes may be smaller than in the SCS. Hence lines 292-300 are overstating the significance of their results.

Similarly lines 287-288 seem to be suggesting that moorings would do a better job than gliders of observing ISWs since they can observe the thermohaline structure at multiple depths simultaneously. Lines 284-291 suggest that using the vertical water velocities as well as the thermohaline data gives you a better chance of detecting ISWs using gliders, but no comparison is made with ship CTD-ADCP surveys. Overall section 4 is a little unconvincing of the advantages of using gliders. (This overlaps with my comment below about section 5.)

**Response:** Moorings conduct the wider application in the observation of ISWs. Unlike mooring array, gliders can adjust the observation path dynamically. This work is a trial of studying the ISWs with glider observation. The characteristics observation of glider is different with moorings, and we conducted the exploratory work of identifying the type of ISWs, which is applicable to glider observation. We hope that more observations can be conducted by gliders to enrich the data *in situ* for ISWs. In the further, the applicability of this method will be verified by experiments in regions where the ISW amplitudes are smaller than in the SCS. It is a pity that we did not do the synchronous observation with ship CTD-ADCP considering the cost of research vessel in this experiment. It is an advantage for gliders to conduct the long-term and continuous observation for obtaining high resolution time series of ISWs.

**Section 5**

**Comment:** The Summary and Discussion section is in fact all summary, and would benefit from some discussion. In the introduction the authors mention several other studies of internal waves/tides from gliders, and thus section 5 could contain some comparison with these previous studies. Similarly the authors could compare their results with other studies observing ISWs in the SCS from other observational platforms. I think they also need to discuss the advantages of using gliders over other platforms that could be used to detect ISWs - and try to find advantages in terms of the quality of science that can be done, not just in terms of what is more costly and time consuming. Otherwise it is not clear what the advantages of this new method are.

**Response:** Thanks for this constructive view.

[Line376-387] "This work is an interesting trial of studying the ISWs with glider observation. Unlike mooring array, gliders can adjust the observation path dynamically. Furthermore, their yo-yo motion allows for

sampling of subsurface features of ISWs which cannot be obtained by satellites. The observation capabilities of glider are similar to the shipboard lowered CTD, but the long-term and continuous observation for obtaining high resolution time series of ISWs poses a challenge to the latter. The glider with characteristic of slow profiling speed is suitable for observing the high-frequency internal waves (Rudnick et al., 2013). However, it is a disadvantage that gliders observe temporal evolution of internal waves considering the weak hold-station capability in the strong current region (Todd, 2017). Besides, although gliders may enable us to obtain high-resolution observation data, there are comparable challenges to estimate the key parameters of ISWs, such as propagation direction and the phase velocities. Direct velocity measurements using current meters or current profilers on gliders might provide a solution to this problem. Future cooperative surveys with a fleet of gliders and multiple observational platforms can be performed to understand the propagation and evolution of ISWs better."

**Quality of written English**

**Comment:** The article would benefit from professional proofreading, as there are numerous grammatical errors, incorrect usage of words, and examples of awkward sentence construction which make the paper considerably less readable. It would also be helpful to focus on using plain English - the meaning is sometimes obscured by the use of unnecessarily elaborate language. For example, line 195 reads "The tendency of theoretical velocity ($w_g$) is coincident with that of vertical velocity ($w_p$)", which I think just means "$w_g$ is broadly similar to $w_p$". I have not noted all such instances of clunky English because that would have meant rewriting nearly every sentence in the article.

**Response:** Those suggestions have been adopted in the revised version. If necessary, we can get professional services to improve the readability after interactive discussion.

[Line 227] "$w_g$ is broadly similar to $w_p$ in Fig.7."

**Inappropriate colour scales**

**Comment:** Figures 8 and 10 use a "rainbow" colour scale, which is not appropriate. It has been widely documented for at least 20 years that rainbow colour scales are not perceptually uniform and can therefore distort interpretation of data and lead to incorrect conclusions. Moreover, figures using rainbow colour scales are not accessible for people with colour blindness, which affects approximately 7% of men and a smaller proportion of women.

Use of rainbow scales is therefore discriminatory. See, for example:

https://tos.org/oceanography/assets/docs/29-3_thyng.pdf

https://www.climate-lab-book.ac.uk/2016/why-rainbow-colour-scales-can-be-misleading/

https://blogs.egu.eu/divisions/gd/2017/08/23/the-rainbow-colour-map/

Figure 8(c) and figure 10 also suffer from having colour bars which do not match the figure. The figure panels show temperature with a colour change every 1 degree, but the colour bars show continuously changing colour. It is important that the colour bars are an exact match to the figures so that readers can easily match colours on the figure to colours in the colour bar.

**Response:** Thank you for the comment on Figure. We enhanced the quality of figures in the revised paper.

[Lines 244-250]

[Figure]

Figure 1 (a) Spatial distribution of the standard deviation ($\sigma$) of $w_c$. The greyscale and contour lines are bathymetry. The black square encloses the area shown in (b), the pink dot-dash lines denote the observing time at intervals of 1 day, as well as the colours representing $\sigma$ for each profile between 4th and 12th August. (b) Zoomed in on the area enclosed in the black square in (a), where the relatively large $\sigma$ in the observation region of Glider-08. (c) Potential temperature obtained by glider-based GPCTD between 4th and 12th August (50 m~500 m). Color shadings and contour lines are isothermals at 1°C intervals. Isothermal surfaces of 10, 15, 20, and 25°C are labeled. The x-axis tick for 4th August starts at the beginning of the 4th .

[Lines 291-292]

[Figure]

Figure 2 Potential temperature recorded during profiles 45-65 (Glider-08). The x-axis tick marks refer to the end of that profile.

**Minor points that apply throughout the paper**

**Comment:**Units: please read the guidance for authors for this journal and correct your use of notation throughout - e.g., m/s should be written m s$^{-1}$. It is not necessary to include a dot between the m and the s$^{-1}$. Also it is conventional to leave a space between the number and the unit, e.g., "1000 m" not "1000m".

Figures: font sizes are too small and hard to read. I had to zoom in to 200% size to read them.

References in text: All references should be referring to the authors. For example, line 114 should read "derived by Ma et al., (2018)" not "derived in Ma et al., (2018)". The latter suggests some very unusual internal surgery done inside the unfortunate Ma!

Use of acronyms and symbols throughout: once you have defined a symbol, you should then always use it. For example, once $w_g$ has been defined you do not need to keep saying "the theoretical vertical velocity ($w_g$) …", just say "$w_g$". Similarly for acronyms.

**Response:** Thank you for the comment on Units, Figure,References, Acronyms and Symbols. We enhance the quality of figure in the revised paper, zoom in the font sizes in the figures, correct the use of notation and acronyms throughout.

**Other comments**

**Comment:**Line 24: "The availability of this method" should read "The effectiveness of this method".

**Response:** [Lines 24-25]: The effectiveness of this method is tested by comparison with a MODIS image. Such an analysis provides a basis for the application of glider in the observation of ISWs.

**Comment:**Lines 43-44: glide speed and horizontal distance travelled are rarely constant and will be affected by ocean conditions and choices made by the glider pilots. Consider giving ranges, or at least averages.

**Response:** [Lines 42-44]: With the aid of wings, the glider flies along a sawtooth trajectory with the typical glide speed of about 0.25 m s$^{-1}$ or horizontal distance of about 20 km per day.

**Comment:** Line 48: remove "at temporal scales" and "on spatial scales".

**Response:** [Lines 46-48]: The low-power consumption and low-speed cruising enable glider to conduct long endurance missions, which can last several months or up to a year and span several hundreds or even thousands of kilometers.

**Comment:** Line 84: could you say a little more about why these gliders perform "best" for profiles deeper than 600 m - and, indeed, in what manner their performance improves for deeper profiles?

**Response:** [Lines 84-86]: The Petrel-II glider is designed for applications to 1500 m depth and perform best for profiles deeper than 600 m, where the buoyancy engine can operate with greater efficiency and glider achieves more higher depth- horizontal distance conversion efficiency.

**Comment:** Line 90-1: What does "According to marine meteorological conditions" mean here? And in what sense did they conduct operations "cooperatively"? Do you just mean "simultaneously"? In fact, could this paragraph from line 89 to line 92 be rewritten as: "ISWs occur more frequently from April to August (Zheng et al., 2007). Since we aimed to take observations during a period of high ISW occurence, the four Petrel-II gliders took simultaneous observations in August (2017). They were deployed to the northeast of Dongsha Atoll and then proceeded southwest and back, along trajectories shown in Fig.1(b)."

**Response:** [Lines 90-93]: Besides, the occurrence frequencies of ISWs also fluctuate significantly from month to month (Zheng et al., 2007). ISWs occur more frequently from April to August. Since we aimed to take observations during a period of high ISW occurence, the four Petrel-II gliders took simultaneous observations in August 2017. They were deployed to the northeast of Dongsha Atoll and then proceeded southwest and back.

The trajectories are shown in Fig.1(b).

**Comment:** Figure 1(b): the caption needs to explain the whole figure, not just the trajectories - it should mention the bathymetry and that areas mentioned in the text are labelled. The colour bar label should read "Depth (m)" not "Depth/m" - the latter implies you are dividing the depth by metres, which makes no sense.
**Response:** [Lines 94-98]:

[Figure]

Figure 3 Glider in the sea trial, map of the South China Sea, glider trajectories, and bathymetry. (a) A Petrel-Ⅱ glider at the surface for communication. (b) The trajectories of 4 gliders deployed in the active region of internal solitary waves, where located in the east of Dongsha Atoll. Red dots indicate starting position of each glider. Thin black curves are isobaths of 500, 1000, 1500, 2000, 2500,3000, 3500 and 4000 m. The black dot represents the Dongsha Atoll.

**Comment:** Lines 100-101: give the expected accuracy of the GPCTD (can normally be found on the manufacturer's data sheets). Also state what pressure sensors are mounted on the gliders, and give the expected accuracy of the pressure sensors.
**Response:** [Lines 104-107]: The accuracy of temperature, conductivity and pressure sampled by GPCTD is $\pm$ 0.002 °C, $\pm$ 0.0003 S m$^{-1}$ and $\pm$0.1 % full scale range, respectively. The pressure sensor with accuracy of $\pm$0.5 % full scale range is mounted on the glider to measure current running depth.

**Comment:** Line 103: "spatial" should be "horizontal".
**Response:** [Lines 107-108]: Therefore, the high sampling frequency and low gliding velocity enable the maximum thermohaline resolution to be 0.15 m in vertical and 2.5 km in horizontal scales.

**Comment:** Table 1 caption: "Glider observations in August 2017. The distance refers to the total horizontal distance travelled over the entire deployment." You don't need to say 'several gliders' in the caption because there's obviously several gliders in the table.
**Response:** [Lines 109-110]: Table 1 Summary of the glider observations in August 2017. The distance refers to the total horizontal distance travelled over the entire deployment.

**Comment:** In lines 145 - 147 the authors refer to the sensitivity of structural parameters to mean theoretical vertical velocity, and then quote numbers which are obviously the sensitivity of mean theoretical vertical velocity to the parameters. Yes, these are simply the inverse of each other, but why not make the terminology consistent with the quoted numbers?

Similarly in lines 152-153. The use of the symbol delta seems inconsistent between figure 2 and the text - and certainly it is unnecessarily confusing. Why not just say "-0.004 cm s$^{-1}$ per percent change in epsilon", for example? (Line 147).

Line 161: describe the specific cost function used - as mentioned earlier, there's more than one cost function which has even been applied to glider flight models.

**Response:** The expression is wrong. We have modified it according to the comment.

[Lines 164-166]: The sensitivity of $V_g$, $\gamma$ and $\varepsilon$ on the mean $w_g$ is 0.0274~0.079 cm s$^{-1}$ mL$^{-1}$, -0.032~-0.033 cm s$^{-1}$ percent change in $\gamma$ and -0.004 cm s$^{-1}$ percent change in $\varepsilon$, respectively, as shown in Fig.2(b). The sensitivity of $\varepsilon$ is lower than others. Thereby, the compressibility coefficient ($\gamma$) and glider volume ($V_g$) have a greater influence on the $w_g$.

[Lines 169-171]: The sensitivity of hydrodynamic coefficients on the mean $w_g$ is given in Fig.3(b). The sensitivity of $C_{D0}$ and $C_L$ is -0.11~-0.07 cm s$^{-1}$ percent change in $C_{D0}$ and -0.068~-0.047 cm s$^{-1}$ percent change in $C_L$, respectively, while the sensitivity of $C_D$ or $C_{L0}$ is close to zero.

**Comment:** Line 163-164: Figure 4 actually shows the parameters against date, not dive number.

**Response:** [Line 182]: Those optimized parameters as a function of date are shown in Fig.4.

**Comment:** Line 169: "The lift coefficient $C_{L0}$ does not change enough to cause a significant change in $w_g$." What matters is not how much all these parameters change relative to each other (especially since their apparent significance is entirely dependent on the y-axis scales you happen to have chosen for figure 4), but how much they would change $w_g$. I suggest you do some simple sensitivity calculations to illustrate this for the reader, something along the lines of: "the mean $w_g$ for the ascent of dive N is M, if using the correct $C_{D0}$ for dive N, but would be calculated as P if using the $C_{D0}$ value from the start of the deployment." Dive N would be towards the end of the deployment.

**Response:** I think that those simple sensitivity calculations are repeated. We have analyzed how much a variation rate of 1% $C_{D0}$, $C_L$ have an effect on the mean $w_g$, which is entirely independent on the y-axis scales of those coefficients. If it is necessary to do some simple sensitivity calculations to illustrate this for the reader, we can follow your suggestion in the latter.

**Comment:** Lines 179-184: start by explaining what a depth-keeping experiment is - remember not all your readers will be familiar with glider capabilities. State the dive number and date so people can see when this was in figure 4. State the values of the original and corrected $V_g$ and gamma. The glider will maintain a constant depth when its density is equal to the density of the surrounding water, which is equivalent to its buoyancy being equal to its weight (not gravity). If you have not previously done so in your rewrite of the early part of section 3, give the equations which allow you to calculate the glider's theoretical density for comparison with the water density. Then figure 6 will make sense. You should also explore how much of the improvement is due to the correction of $V_g$ and how much is due to the correction of gamma, rather than just showing the combined effect.

**Response:** [Lines 199-207]: The glider will maintain a constant depth when its density is equal to the density of the surrounding water, which is equivalent to its buoyancy $F_b$ being equal to its weight $F_b$ in the Eq.(1). We conducted a depth-keeping experiment to realize this motion by setting appropriate buoyancy to validate the structural parameters. The correction of $V_g$ changes 6.88 mL relative to the original value, but the change of $\gamma$

is almost zero. The reason is that the glider completes 26 profiles in the 3 days after deployment. As can be clearly seen in Fig.4(d), almost no change occurs in the $\gamma$ for a relatively shorter running duration. The depth-keeping experiment and simulations with original and corrected structural parameters are shown in Fig.6. The experiment lasted for about half an hour, and the glider held depth at 848±1 m. Substituting the original and corrected parameters into the buoyancy model yields the depth simulations, and the error of depth decreases from 24 m to 12 m when corrected parameters are adopted.

**Comment:** Line 188: $w_p$ is not relative to the water, $w_g$ is.

**Response:** [Line 211]: The actual vertical velocity $w_p$ is obtained by the time rate of change of pressure measured by the CTD. $w_g$ is the theoretical velocity inferred from flight model.

**Comment:** Lines 189-90: Does "excellent time-frequency characteristic" mean "high sampling frequency"?

**Response:** No, the excellent time-frequency characteristic is the feature for the wavelet signal denoising. We adopted the wavelet signal denoising is applied to restrain high frequency noises from the pressure transducer signal.

**Comment:** Figure 7: The x-axes of panel a and panel b are not aligned. Panel b contains slightly more than 4 dives, panel a contains nearly 6 dives. In panel a, you do not need two y- axes because they're identical. Just label the axis as 'velocity (m s$^{-1}$)' and then the legend explains which is which. The caption needs to explain panel a and panel b separately.

The caption for panel b needs to explain what is meant by 'shift', since this is mentioned nowhere in the text (I think you mean battery position along the long axis of the glider, relative to the central position?), and that theta is pitch. The caption should also state that this is a representative 12 hour period, and you should make sure the main text reflects this - e.g., line 191 should say something like "a 12-hour period, representative of the entire dataset, is shown in Fig 7" not "as shown in Fig. 7". The second sentence in the caption: "Given the unsteady motion during the eccentric battery pack shifting or rotating, and variable buoyancy engine working, the velocities in those moment are excluded" - this needs to go in the main text with more detail. Are these times excluded while you're optimising the flight model parameters, or while you're examining the resultant water velocities for ISWs, or both? Do you just exclude the exact times when the glider is pumping/pitching/rolling or do you also exclude a little more data after those times to allow the flight to stabilise? The glider will not respond instantly to changes in battery position.

**Response:** We have modified it according to the comment.

[Lines 215-226]: Besides, the glider will not respond instantly to pitch adjustment by moving a battery pack. The pitch after applying a smoothing window (60 s) is used to calculate the vertical water velocity. Based on the vertical velocity estimation method and optimized parameters, the water and glider vertical velocities are achieved. Given the unsteady motion during the eccentric battery pack shifting or rotating, and variable buoyancy engine working, those factors cause the spikes on the vertical speed of glider. The velocities in those moment are excluded and not be used to examine the resultant water velocities for ISWs. The velocity, pitch, shift, depth during a 12-hour period, representative of the entire dataset, is shown in Fig.7.

[Figure]

Figure 4 the data from a 12-hour period, representative of the entire dataset Given the unsteady motion during the eccentric battery pack shifting or rotating, and variable buoyancy engine working, the velocities in those moment are excluded. The shift means the battery position along the long axis of the glider, relative to the central position. (a) Vertical velocity versus time. (b) Pitch, shift, depth versus time.

**Comment:** Lines 195-203: You say Merckelbach et al. proposed a robust approach to estimate vertical water velocities, but you've just spent the last 4 pages telling us how you estimate the vertical water velocities. Do you mean Merckelbach et al. proposed a method to estimate the *errors* on vertical water velocities? If yes, you still need to summarise their method here, because at the moment there is no explanation for the stated inaccuracy of 4mm/s.

Line 199: What is the significance of mean water velocities for 3-day periods? Why 3 days and not a mean over the whole time series?

**Response:** Merckelbach et al. proposed a method to estimate the errors on vertical water velocities.

[Lines 229-230]: A robust approach to estimate the error of vertical water velocities is proposed in the reference (Merckelbach et al., 2010).

"Although lacking an independent measurement of the vertical velocities, the previous three subsections have presented robust but indirect evidence of the calculated vertical water velocities to be reasonably accurate. But, how accurate is ''reasonable''? To quantify the accuracy, we first look at the mean values. The mean values of the data shown in the histograms of Fig. 15 are found to be -1.8 and -2.5 mm s$^{-1}$. The mean value of the vertical velocity of the whole record amounts to 0.1 mm s$^{-1}$. After the cost-function minimization, the mean values for 3-day periods are $0.2 \pm 4$ mm s$^{-1}$. Calculating the mean values per depth bins of 50 m, the variation is typically less than 0.1 mm s$^{-1}$. Therefore, it seems that the systematic error in the vertical velocity is approximately $\pm 4$ mm s$^{-1}$." (Merckelbach et al,2009)

We adopt the same method to estimate the accuracy of vertical water velocities. Seaglider adopt this method to estimate the inaccuracy. (Frajka-Williams et al., 2011)

**Comment:** Thoughout Section 4, you appear to be using profile number for dive number, in both the text and the figures. If you wish to refer to each profile separately, rather than referring to "Profile-47 in the diving phase", which gets rather clumsy, then simply explain that you are numbering them profile by profile. So, for example, the descent of dive 1 becomes profile 1, the ascent of dive 1 becomes profile 2, the descent of dive 2 becomes profile 3, etc. Thus you can use 'dive' when you mean the descent and the ascent together, and

'profile' for the descents and ascents separately.

**Response:** It is an obscure description for profile. We modified throughout the text according to the comment. We use the profile to refer to descents and ascents of glider. We utilized the descents instead of dive throughout the text.

**Comment:** Also throughout section 4, it is common to use the Greek letter sigma for standard deviation, rather than "std".

**Response:** Thank you for the suggestion. Modified throughout the text according to the comment.

[Lines 241-242]: The spatial distribution of the standard deviation ($\sigma$) of $w_c$ during each profile in this region from 4$^{th}$ to 16$^{th}$ August is mapped out as shown in Fig.8.

**Comment:** Lines 205-206: Remove the first two sentences of this paragraph, they simply repeat information presented in section 3.

**Response:** We have modified it according to the comment.

**Comment:** Line 207: Give a more complete explanation of your chosen depth range, e.g., "Depths above 50 m are excluded because …, depths below 500 m are excluded because …". Also remove the word "therefore".

**Response:** [Lines 237-241]: We analyze the vertical water velocities within the common depth from 50 m to 500 m during the steady gliding motion. Depths above 50 m are excluded because the variable buoyancy engine works at the surface and glider can realize stable flight after that depths. The four gliders have a non-uniform depth in the sea trial, so the data above common depths 500 m are taken for the further analyses.

**Comment:** Figure 8: do not use the same colour scale for two different properties in the same figure. In other words, potential temperature should not use the same colour scale as sigma. On panel c, I suggest you change the axes tickmarks to come out from the axis instead of in, because they're completely invisible at the moment. The x-axis label would be better as "day in August 2017 (UTC)".

Figure 8 caption: the caption needs to describe the whole figure, not just the aspects to which you wish to draw the readers' attention. So the caption for panel a needs to explain that the greyscale and contour lines are bathymetry, the black square encloses the area shown in panel b, the pink dot-dash lines denote the observing time at intervals of 1 day (this should not be in the main text)- as well as the colours representing sigma for each dive between 4$^{th}$ and 12$^{th}$ August. For panel b, the caption should simply say 'as panel a, zoomed in on the area enclosed in the black square in panel a'. Avoid words such as 'dramatic', this is unscientific. Panel c's caption needs to explain the x-axis ticklabels a little more - e.g., is the tick for 6$^{th}$ August at midnight at the beginning of the 6$^{th}$, or at noon on the 6$^{th}$, or at midnight at the end of the 6$^{th}$? All are possible.

**Response:** We replot the Figure 8 and modified the caption in the revised paper according to the comment.

**Comment:** Line 217: do you mean sigma increases to the east or to the west? And what about the obvious increase to the south?

**Response:** Yes. [Lines 253-255]: This variation indirectly reflects that the activity of vertical water increases gradually to the west and to the south, and such distribution coincides with fluctuation of isotherms in the 4 temperature transections measured by glider-based CTD.

**Comment:** Lines 223-225: "sigma increases to 5.06cm/s (20°54.08′N, 117°49.23′E, dive NN), 4.83cm/s (20°50.02′N, 117°46.67′E, dive NN), and 4.22cm/s (20°39.48′N, 117°40.42′E,dive NN), and these values are considerably larger than the average." Line 229: a velocity cannot be 'sharp'.

**Response:** [Lines 259-261]: The $\sigma$ increases to 5.06 cm s$^{-1}$ (20°54.08′N, 117°49.23′E), 4.83 cm s$^{-1}$ (20°50.02′N, 117°46.67′E), and 4.22 cm s$^{-1}$ (20°39.48′N, 117°40.42′E), respectively, and these values are considerably larger than the average.

**Comment:** Line 237-8: "further analyze the profiles with unusually large sigma (Fig 8(b))."

**Response:** [Line 273]: This, together with characteristic of ISWs, can be used to further analyze the profiles with unusually large $\sigma$ (Fig 8(b))

**Comment:** Figure 9 caption: "Depth, w$_p$ and w$_c$ measured by Glider-08 on the 9$^{th}$ and 10$^{th}$ of August." Figure 11's caption should be altered similarly.

Figure 9, profiles 46-48 is labelled as commencing at 6pm on August 9$^{th}$. Figure 11, profiles 61 to 65, is labelled as commencing just before midnight on August 9$^{th}$. This seems inconsistent.

**Response:** Thank you for underlining this deficiency. This mistake is corrected. Modified Figure 9's and 11's caption according to the comment.

[Line 277]: Figure 5 Depth, $w_p$ and $w_c$ measured by Glider-08 on the 9th and 10$^{th}$ August

[Line 306]: Figure 6 Depth, $w_p$ and $w_c$ measured by Glider-08 on the 12$^{th}$ August

**Comment:** Lines 245-6: "The strong upwelling forces the glider to change its predefined movement direction from downward to upward at 22:20 UTC." This reads as though the glider is somehow forced to pitch nose up and pump oil to the external reservoir to try to fly upwards - which of course is not what you mean. I suggest "The strong upwelling around 22:20 UTC induces a temporary reversal in w$_p$ from downward to upward."

**Response:** [Lines 281-283]: The strong upwelling around 22:00 UTC induces a temporary reversal in $w_p$ from downward to upward. Those rapid vertical water velocity fluctuation leads to the profile with unusually large $\sigma$ in the Fig.8(b).

**Comment:** Figure 10 caption: "Potential temperature recorded during profiles 45-65 (Glider-08)." What are the white triangles along the upper x-axis? As with the day numbers in figure 8(c), you need to explain whether the x-axis tickmarks refer to the start, middle or end of that dive/profile.

Figure 10 compared with lines 252-3: on figure 10 it looks very much as though the downward displacement of isotherms occurs during dive/profile 46, not 47.

**Response:** We replot the Figure 10. There is 30°C isotherms along the upper x-axis.

[Line 292]: Figure 7 Potential temperature recorded during profiles 45-65 (Glider-08). The x-axis tick marks refer to the end of that profile.

[Lines 289-290]: The passage of ISWs captured by Profile-47 induced sunken displacements of isothermals, and this phenomenon coincides with characteristics of a depression ISW.

**Comment:** Line 257: "making the thermocline sunk to nearly 300m depth." You have not defined a specific thermocline. On the figure it appears to be the 13 degree isotherm that sinks to nearly 300 m? Assuming the x-axis tickmarks refer to the start of each profile, there seem to be depressed isotherms between approximately

170 - 350 m during profile 60, and it's only the 11 degree isotherm that is depressed during profile 61.

**Response:** [Lines 293-294]: As is clear in Fig.10, it seems that a small depression wave arrived in profile-61, making the 13℃ isotherms sunk to nearly 300m depth.

**Comment:** Lines 270-279: if ISWs can be detected from satellite images, why do we need to use gliders at all? What extra information is gained by using gliders? (This might actually be something to include in the summary and discussion section rather than here.)

**Response:** We gave a summary and discussion in the last section.

[Lines 369-387]: This paper investigates the activity of internal waves in the eastern area of Dongsha Atoll. The approach and accuracy to estimation of vertical water velocity reported by previous authors (Merckelbach et al., 2010; Frajka-Williams et al., 2011; Rudnick et al., 2013) is not be discussed in detail herein, because those methods have been proven effective and applied in the research of internal waves. We conducted the exploratory work of addressing the question of whether gliders can detect ISWs and identifying the type of ISWs, which is applicable to glider observation, and have been demonstrated around the Dongsha Atoll in the SCS. In the further, the applicability of this method will be verified by experiments in regions where the ISW amplitudes are smaller than in the SCS.

This work is an interesting trial of studying the ISWs with glider observation. Unlike mooring array, gliders can adjust the observation path dynamically. Furthermore, their yo-yo motion allows for sampling of subsurface features of ISWs which cannot be obtained by satellites. The observation capabilities of glider are similar to the shipboard lowered CTD, but the long-term and continuous observation for obtaining high resolution time series of ISWs poses a challenge to the latter. The glider with characteristic of slow profiling speed is suitable for observing the high-frequency internal waves (Rudnick et al., 2013). However, it is a disadvantage that gliders observe temporal evolution of internal waves considering the weak hold-station capability in the strong current region (Todd, 2017). Besides, although gliders may enable us to obtain high-resolution observation data, there are comparable challenges to estimate the key parameters of ISWs, such as propagation direction and the phase velocities. Direct velocity measurements using current meters or current profilers on gliders might provide a solution to this problem. Future cooperative surveys with a fleet of gliders and multiple observational platforms can be performed to understand the propagation and evolution of ISWs better.

**Comment:** Data availability: it is stated that the data from the field experiments is only available on request from the authors, with no detailed explanation of why the data is not available from a public data repository. This does not seem adequate to meet Ocean Science's data availability requirements.

**Response:** The experiment was carried out in collaboration with other universities and institute. After consulting with the partners, we can share the data with the researchers who are interested in the experiment and ISWs in the South China Sea privately. We are so terribly sorry that we cannot upload the data to a public data repository.

Before submit the manuscript to the Ocean Science, we noticed that some research articles( e.g. Stranne et al., 2018) in the Ocean Science are allowed to share the data supporting the figures and text in paper upon request from the corresponding author.

**References**

Fer, I., A.K. Peterson, J.E. Ullgren, 2014, Microstructure Measurements from an Underwater Glider in the Turbulent Faroe Bank Channel Overflow, Journal of Atmospheric and Oceanic Technology.

Liang, X., M. Spall, C. Wunsch, 2017, Global Ocean Vertical Velocity From a Dynamically Consistent Ocean State Estimate, JGR Oceans.

Frajka-Williams, E., Eriksen, C.C., Rhines, P.B., Harcourt, R.R., 2011. Determining Vertical Water Velocities From Seaglider. J. Atmos. Oceanic Technol. 28(12), 1641-1656.

Marshall, J., Schott, F., 1999. Open-Ocean Convection: Observations, Theory, and Models. Reviews of Geophysics. 37(1), 1-64.

Merckelbach, L., Smeed, D., Griffiths, G., 2010. Vertical Water Velocities From Underwater Gliders. J. Atmos. Oceanic Technol. 27(3), 547-563.

---

## Author Comment (AC2)

**Comment on os-2021-29**

Anonymous Referee #2

Referee comment on "The inference of internal solitary waves in the northern South China Sea from data acquired by underwater gliders" by Wei Ma et al., Ocean Sci. Discuss., https://doi.org/10.5194/os-2021-29-RC2, 2021

The authors did an in situ experiment and demonstrated that the observations using gliders can show the existence of ISWs. They have taken a first step in addressing the question of whether gliders can detect ISWs. I have only some minor comments on the manuscript.

Dear Referee #2:

We would like to thank you for your efforts in reviewing our manuscript and providing many helpful comments and suggestions. Those comments are all valuable and very helpful for revising and improving our paper, as well as the important guiding significance to our researches. We have studied comments very carefully. Based on comments and suggestions, we have revised the manuscript accordingly. The details are explained as follows.

Sincerely yours,

Wei Ma

On behalf of all authors.

**Comment:** Line 29: The satellite observations suggest that the largest distance travelled by ISWs is on the order of 600 kilometers, occurring in the SCS. Thus, I suggest to change 'thousands of kilometers' to 'hundreds of kilometers'.

**Response:** Thank you for underlining this deficiency.

[Lines 29-30]: Internal solitary waves (ISWs) are ubiquitous features in the ocean, and they can propagate over hundreds of kilometers from the generation site with unusually strong currents,

**Comment:** Line 32: It is Liu et al. (1985, JPO) that clarifies the weakly nonlinear internal wave theory. It is more appropriate to add this reference.

**Response:** We added "Liu et al. (1985, JPO)" on Line 32

[Line 32]To well understand the ISWs, several theories have been proposed (Liu et al., 1985; Cai et al., 2014).

**Comment:** Line 82ï¼□ISWs are not generated within Luzon Strait. They originate from Luzon Strait.

**Response:** [Lines 81-82]: Most ISWs in the northern South China Sea are originated from Luzon Strait, and propagate westward (Simmons et al., 2011).

**Comment:** Figure 9 : Can we approximately treat the trough-peak oscillation in the No.47 profile as that induced by a same ISW?

**Response:** In my opinion, we can approximately treat the trough-peak oscillation in the No.47 profile as that induced by a same ISW, because the vertical water velocity of adjacent profiles did not experience the continuous and violent fluctuation.